# Substantial loss of trawlable biomass and lack of recovery in a marine ecosystem
Jacob Burbank [1,2] ✉, Nicolas Rolland[1,2], Jenni L. McDermid[1], François Turcotte[1], Tyler D. Tunney[1], Daniel Ricard[1] & François-Étienne Sylvain[1]

Profound changes in species assemblages are occurring in marine ecosystems worldwide and are essential to document. Here we use 51 years (1971–2021) of fishery-independent data from a standardized bottom-trawl research vessel survey (6440 independent fishing locations) in the southern Gulf of St. Lawrence covering 70,091 km² to evaluate trends in marine community structure and trawlable biomass across 122 fish and crustacean taxa. Survey data indicate a substantial decline in biomass and increase in turnover for taxa susceptible to bottom-trawl fishing gear in the southern Gulf of St. Lawrence marine ecosystem that corresponds with the reduction of several predatory fish and a major regime shift around the early 1990's. Unlike other marine regime shift examples, we observe a substantial net loss of trawlable biomass in the community, with limited compensatory response in small fish and crustacean biomass over nearly 30 years following the depletion of predatory groundfish. Overall, this unique case of reduced biomass and shift in community structure highlights the importance of maintaining and analyzing fishery-independent surveys over extended time series. Such information is vital to assessing the state of marine ecosystems and developing plans for recovery, as we face a future of untold challenges in managing marine ecosystems worldwide.

Marine species fluctuate in response to a number of factors including climate, ecosystem factors and anthropogenic pressures such as pollution and commercial and recreational fishing. Since the development of industrial fishing, fisheries have contributed to significant and sustained declines of several marine species in areas across the globe[1,2]. Such declines in marine stocks and a move towards sustainable fisheries have been a focal point of fisheries research and have received attention from media and society for the past few decades[3]. Yet, despite reductions in fishing activity and the implementation of fishing moratoria for some stocks, declines in both biomass and body size of fish populations within some marine communities persist[4].

Declines in biomass of fish populations have been documented worldwide. However, studies have often focused only on commercially important fish[5–7], typically using fishery-dependent data[8] and potentially failing to examine and quantify trends across the fish community. Other studies that reach beyond fishery data have often been based on complex ecosystem models with a wide range of assumptions[2,9]. Studies examining loss of biomass or shifts in structure of marine communities, including non-commercial species, using long-term, high-resolution fishery-independent survey data have been valuable in conceptualizing ecosystem change along the northeast US continental shelf[10–12] but are lacking for the southern Gulf of St. Lawrence (sGSL). Declines in marine populations can be difficult to conceptualize, particularly due to a common phenomenon known as the 'shifting baseline syndrome'[13], where each new generation of fisheries scientists set a baseline for stocks at the beginning of their careers, and evaluate changes relative to this baseline[13,14]. This new baseline often fails to account for changes that occurred in the previous generation, thereby dampening the perception of population decline through time. This can result in subsequent generations accepting lower stock sizes as 'acceptable' and sustainable, and can lead to the development of unambitious recovery targets as the baseline shifts through time[15]. Using truncated time series and defining reference points from these often leads to a skewed perception of stock status most often by missing important historical biomass[16]. However, an important challenge faced by investigations of population and community dynamics that span generations is to address changes in scientific survey designs, fishing gears and technological advances in fishing approaches to ensure long-term datasets are comparable through time[17,18].

Nevertheless, true declines in fish stocks can have detrimental impacts on marine communities and their supporting ecosystems[1,3,19]. Stocks targeted by commercial fisheries are currently or were historically abundant

[1]Fisheries and Oceans Canada, Gulf Fisheries Centre, Moncton, NB, Canada. [2]These authors contributed equally: Jacob Burbank, Nicolas Rolland. ✉e-mail: Jacob.Burbank@dfo-mpo.gc.ca

within an ecosystem[1,20], and thus often have substantial ecological value[1,20]. Therefore, when the abundance of these stocks experiences rapid declines due to exploitation, the overall biomass and composition of the community can change significantly. Such declines of important species can shift marine communities to alternate states, impact trophic structure, and reduce overall biomass[19]. Coupled with the effects of changing ecosystem conditions and ongoing climate change, community structure and function can shift further, resulting in important changes in biomass, abundance and size composition of fish populations throughout marine systems.

It is generally accepted that fishing tends to cause more rapid declines in the biomass of larger, longer-lived, demersal, and generally piscivorous species compared to smaller, shorter-lived forage fish[4,8,19,21]. In some ecosystems, this has allowed smaller or unexploited species to thrive in response to the absence of predators, competitors, or following favourable changes in prey base or other important ecosystem factors, thereby resulting in the recovery of biomass along with changes in community structure and ecosystem function[22,23]. In some rarer cases, these community-level changes may be short-lived as trends can quickly reverse with changes in stressors such as reductions or cessation of fishing activity through the implementation of lower quotas, no-take reserves, or fishery closures[24,25]. Alternatively, changes in trophic structure and biomass can be persistent as long-lived exploited species often take decades to recover, or fail to recover, as observed with stocks of Atlantic cod (*Gadus morhua*) and White hake (*Urophycis tenuis*) along the Atlantic coast of Canada[7,26,27]. Furthermore, unfavorable changes in environmental conditions and altered interspecific interactions such as increased predation pressure can suppress the recovery of previously overexploited species and alter ecosystem dynamics[27,28]. Given the complexity of interspecies interactions, it is valuable to use available data from fishery-independent multispecies research surveys to examine and quantify changes in trawlable biomass and community structure through time to better understand potential impacts of anthropogenic activity and climate change on marine ecosystems. Examining community-level shifts provides a different perspective compared to more traditional species-specific analyses and may help to better articulate ecosystem changes over time.

Here we examine data from an uninterrupted 51-year time series from a fishery-independent multispecies bottom-trawl survey conducted annually during September in the sGSL by Fisheries and Oceans Canada (Fig. 1). The survey is used to evaluate temporal trends in the biomass and abundance of fish and crustaceans. Here, we evaluate the temporal changes in the biomass and total number of fish captured per tow. We also examine the relative biomass of each species throughout the time series to identify regime shifts in community structure and relate them to shifts in biomass. Overall, this study highlights significant reductions in trawlable biomass and shifts in community structure over an extended period of time, improving our understanding of the dynamics of an important marine ecosystem and paving the way for similar studies in marine ecosystems worldwide that currently have long-term fishery-independent bottom trawl surveys.

## Results
### Grouped biomass and abundance indices
The biomass of all fish taxa averaged 175 kg per tow over the time series. This index was at its highest between 1974 and 1991 with a mean value of 284 kg per tow and maximums observed in 1981 and 1988 at 422 and 369 kg per tow, respectively (Fig. 2). In 1992, biomass decreased sharply to 162 kg per tow followed by a slow sustained decline to a mean value of 110 kg per tow. The lowest biomass observed in the time series was 62 kg per tow in 2019. In contrast, abundance, expressed in number per tow, revealed a relatively flat pattern with a mean number of 824 specimens per tow from the beginning of the time series until 2009. Abundance then rapidly increased to a mean value of 2178 specimens per tow in 2017 before returning to values averaging 1232 specimens per tow in years 2018 to 2021 (Fig. 2).

Biomass and abundance indices of the commercial fish group (Fig. 2) show similar trends to those for all taxa combined. This group includes important commercial species such as Atlantic cod, Atlantic herring (*Clupea harengus*), Atlantic mackerel (*Scomber scombrus*), Atlantic halibut (*Hippoglossus hippoglossus*), American plaice (*Hippoglossoides platessoides*), Redfish (*Sebastes* spp.), and several flounder species. These species are or were part of major commercial fisheries within the sGSL over the past 50 years (see Supplementary Table 1 for the list of fish species in the group). From 1971 to 2015, the 21 commercial species represented an average of 87% of the total biomass of the 122 taxa caught during the trawl survey with a high of 97% in 2007 and a low of 73% in 1992 (Fig. 2). However, this

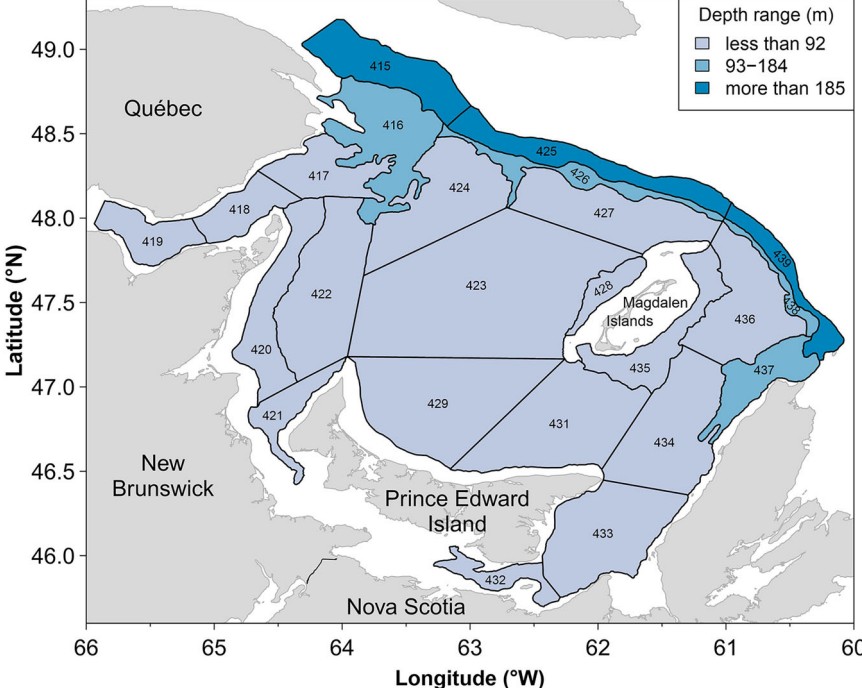

**Fig. 1 | Map of the strata sampled during the yearly research vessel survey in the southern Gulf of St. Lawrence.** The shades of blue correspond with depth ranges.

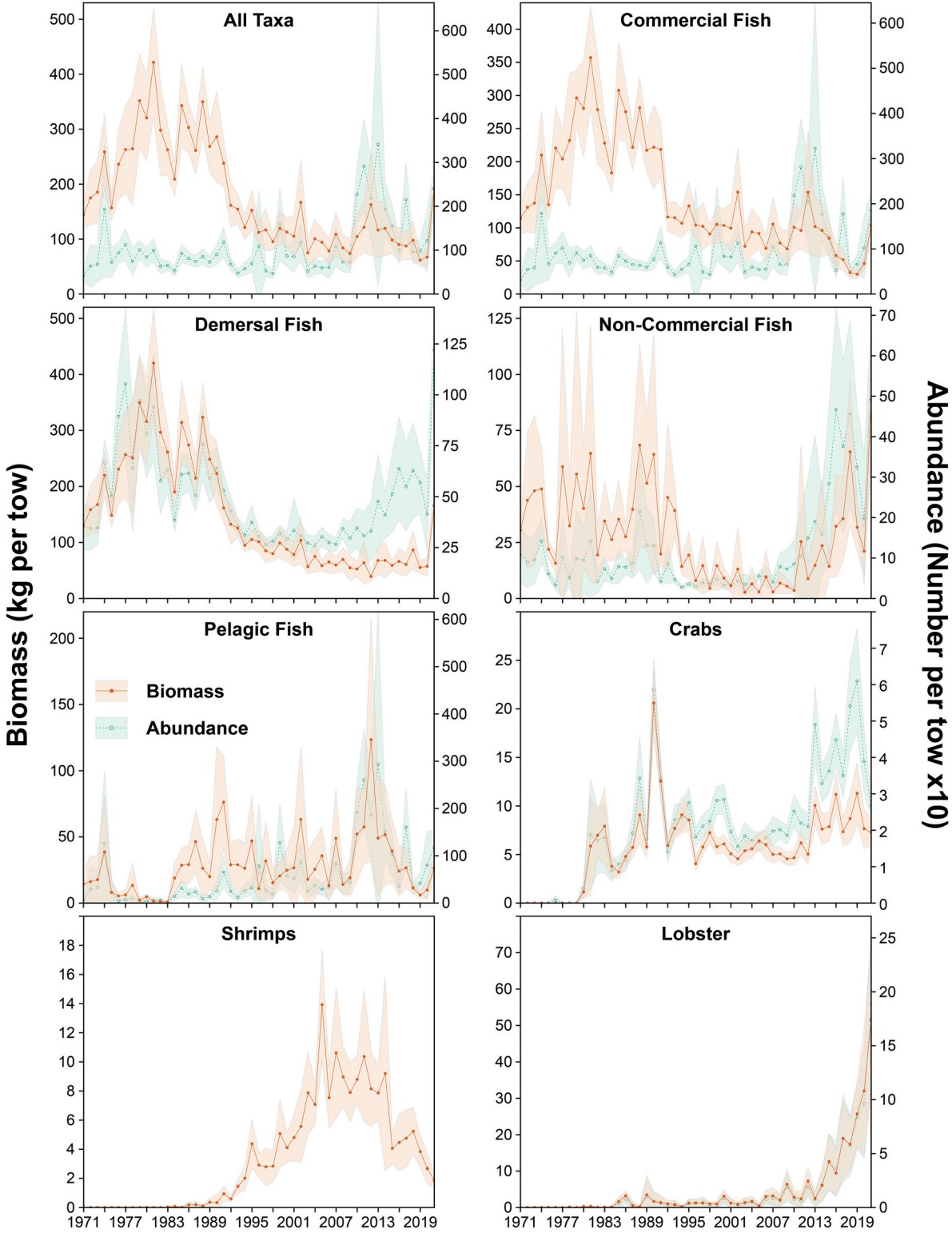

**Fig. 2 | Temporal trends in biomass and abundance.** Temporal trends in the biomass of fish and crustaceans expressed as kg per tow (orange) and abundance expressed as number per tow (blue) from the fishery independent multispecies research vessel survey conducted in the southern Gulf of St. Lawrence. The shading indicates the 95% confidence interval of biomass and abundance estimates. Taxa included in each group are found in Supplementary Table 1.

proportion has decreased substantially since 2015, reaching a value of 44% in 2021 after a record low minimum of 32% in 2018 (Fig. 2). Similar proportions and trends were observed when examining the number of individuals per tow. The spike in the abundance of commercial species observed from 2010 to 2014 is largely driven by an increase in the abundance of small-bodied pelagic species (mostly Atlantic herring), which has since declined. Since 2019, there has also been a slight increase in biomass and abundance, largely driven by a recruitment pulse of recovering Redfish (Fig. 3).

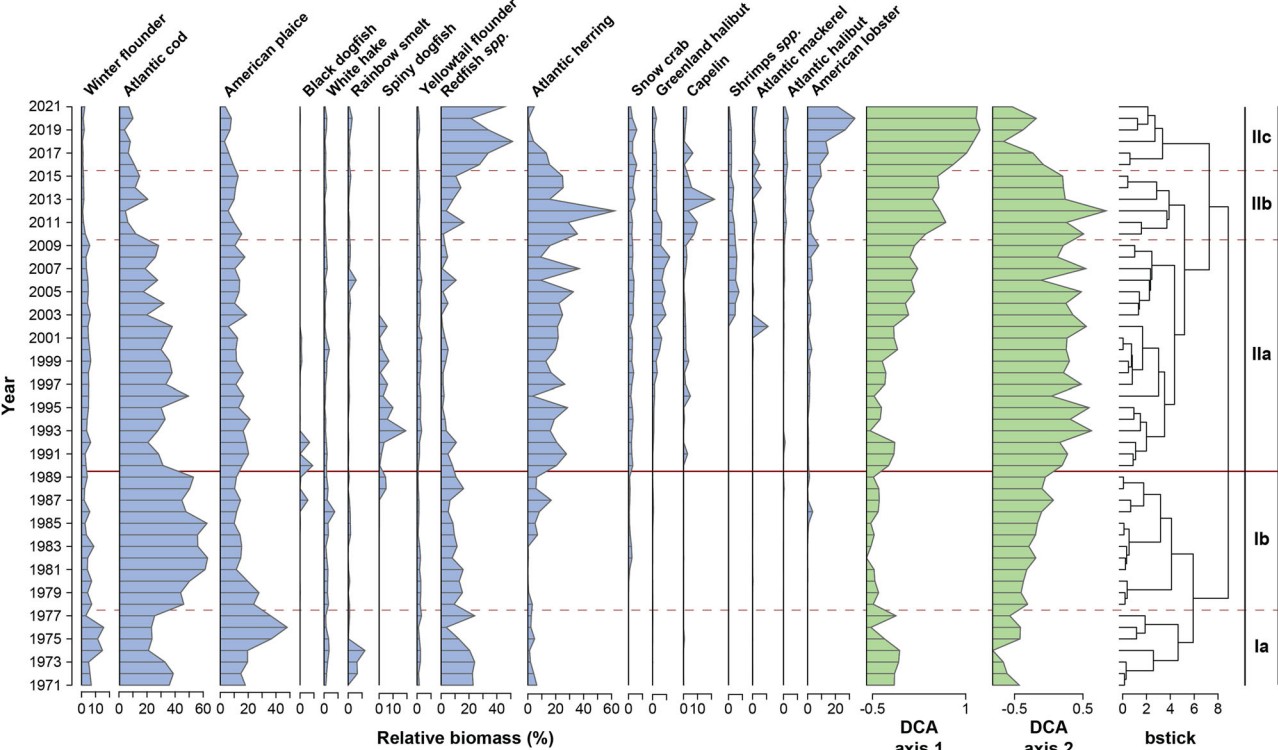

**Fig. 3 | Relative biomass of key species.** The relative biomass of 17 selected taxa from 1971–2021 (filled blue), the DCA axes scores (filled green) representing rates of turnover in the community (axis-1) and uninterpretable variation (axis-2), and the taxa groups identified by the cluster analysis. The horizontal red line delineates the two main time periods (I and II) identified by the CONISS algorithm using the broken-stick model (bstick) and represents a regime shift in the marine community in the sGSL. Three dashed red lines delineate subperiods (Ia, Ib, IIa, IIb, and IIc) within the main time periods (I and II) where the patterns of dominance in community member biomass were found to cluster.

The demersal fish species biomass (Fig. 2) follows a similar trend to that observed for the group containing all taxa, with a mean value of 147 kg per tow over the time series. The period between 1974 and 1991 averaged 262 kg per tow. In 1992, biomass rapidly declined to a mean value of 78 kg per tow with minima observed in 2012 (40 kg per tow) and 2019 (56 kg per tow). Abundance (Fig. 2) shows a similar trend to biomass for the pre-2007 period, with a maximum abundance averaging 687 specimens per tow observed between 1974 and 1991. A sharp decrease then began in 1992, reaching an average value of 311 specimens per tow. In contrast to biomass, abundance began to increase in 2007 and reached a mean value of 583 specimens per tow between 2016 and 2021, even though biomass remained fairly stable. This trend in specimens per tow over the last decade is similar to the one described for the non-commercial fish species (Fig. 2).

Non-commercial fish species biomass and abundance indices (Fig. 2) fluctuated around mean values of 28 kg and 118 individuals per tow over the time series. The pre-1995 period showed the highest inter-annual variability with maximums observed between 1977 and 1981 (mean value of 51 kg and 99 specimens per tow) and 1988–1990 (mean value of 62 kg and 160 specimens per tow). This period is followed by two relatively stable decades with a mean value of 7 kg and 42 specimens per tow until 2008 when the number of specimens per tow showed a rapid increase to reach average values of 242 between 2008 and 2021 and a maximum of 467 in 2016 (Fig. 2). This increase in abundance was observed in the biomass approximately 4 years later, in 2011. Biomass rapidly rose to a mean value of 31 kg per tow between the years 2011 and 2021.

The biomass and abundance indices for pelagic fish species (Fig. 2) were relatively low from 1971 to 1983, with an average of 10 kg and 111 specimens per tow. However, from 1984 to 2009 the indices fluctuated with high inter-annual variability around a mean value of 31 kg and 404 specimens per tow. From 2010 to 2015 these indices present a similar pattern as those observed with all species and the commercial species with

an increase to reach mean values of 62 kg and 1748 specimens per tow from 2010 to 2014, before declining to a minimum of 5 kg per tow in 2019 and 328 specimens per tow in 2018. In the most recent years (2018–2021) the biomass has remained low at an average of 13 kg per tow, while abundance has increased to an average of 654 individuals per year.

The biomass and abundance of crab species (Fig. 2) were at their lowest until 1980 when values began to increase to reach a mean value of 5 kg and 34 specimens per tow between 1986 and 2012. From 2013 to 2021, the indices have fluctuated with an increasing trend at mean value of 6 kg and 65 specimens, particularly in abundance which reached a maximum value of 97 specimens per tow in 2019.

Shrimps biomass was at their lowest at the beginning of the time series (Fig. 2) but began to increase in the 1990s, reaching a mean value of 14 kg per tow in 2005. The biomass then fluctuated around a mean of 9 kg per tow for approximately a decade before decreasing rapidly to a mean of 4 kg per tow in 2014. The biomass has continued to decline in recent years, reaching 2 kg per tow in 2021.

Similarly to the crab and shrimp indices, lobster biomass was very low at the beginning of the time series (Fig. 2), but increased to a mean value of 2 kg and 6 specimens per tow between 1985 and 2013. From this point, the indices have increased to reach a maximum value of 40 kg and 140 specimens per tow in 2021.

## Relative biomass index

The time series of individual species biomass relative to the total biomass of all species was divided into two statistically distinct time periods by the CONISS algorithm (Fig. 3 and Supplementary Fig. 1). The first period covers the years 1971 to 1990 and was subdivided into two subperiods Ia and Ib with a separation line in 1978. The second time period spans the years 1990 to 2021, with a first subperiod (IIa) up to 2010, a second subperiod (IIb) up to 2016 and a third subperiod (IIc) to 2021. The first axis of the

https://doi.org/10.1038/s42003-025-08240-3                                                          **Article**

Detrended correspondence analysis (DCA) analysis shows low inter-annual variability and turnover rates from 1971 to 1990, which characterized a more stable community structure. This period was followed by a period of relatively rapid and constant increase in the DCA axis 1 scores from 1990 up to 2021, revealing a more variable community structure with higher turnover in the species composition within the sGSL (Fig. 3).

**Time period I (1971–1990).** In the first time period, the sGSL fish community was dominated by Atlantic cod, which increased from a mean relative biomass of 31% in subperiod Ia to a mean value of 51% in subperiod Ib. The subperiod Ia is characterized by a higher relative biomass of flat fish, including American plaice (27%) and the inshore Winter flounder (*Pseudopleuronectes americanus*) (9%). The other most abundant species in subperiod Ia was Redfish with a mean relative biomass of 19%. While transitioning into subperiod Ib, American plaice, Winter flounder and Redfish relative biomass decreased and stabilized to mean values of 15%, 5% and 11%, respectively (Fig. 3). The demersal White hake was encountered at low but steady levels (<3%) throughout this period. Atlantic herring was found in subperiod Ia at low levels (3%), then was almost absent in the first half of subperiod Ib (~1%) but quickly increased in the second half of Ib to reach a mean value of 6%. This period is also characterized by the presence of Rainbow smelt (*Osmerus mordax*) in the early 1970s with a maximum value of 5% in 1974, and short-lived presence of Black dogfish (*Centroscyllium fabricii*) and Spiny dogfish (*Squalus acanthias*) later in that period. Finally, American lobster (*Homarus americanus*) was encountered at very low (<1%) levels throughout this period, with only a pulse in 1986 of almost 4%, and Snow Crab (*Chionoecetes opilio*) was only found in subperiod Ib with a mean value of 1% and a maximum of 6% in 1990.

**Time period II (1990–2021).** The second period differs from the first mainly because it is characterized by decreased biomass of many demersal taxa (Fig. 3). White hake was virtually absent, and Atlantic cod fluctuated around a mean value of 29% in subperiod IIa before decreasing to a mean value of 7% in subperiod IIc. American plaice (15% down to 6%) and Winter flounder (5% down to 1%) followed a similar trend. Atlantic herring relative biomass averaged 21% in the subperiod IIa before increasing to a mean value of 31% in subperiod IIb with a major peak in 2012 at 64% before plummeting to a mean value of 9% in the most recent years (IIc).

The decrease in biomass of most of the important commercial species of the system was partially counterbalanced by a rapid increase of both Redfish and American lobster. With a relative biomass of less than 0.05% in the first period, Capelin (*Mallotus villosus*) and Atlantic mackerel, two pelagic species, became more abundant in the second period with a mean relative biomass of 6% (IIb) and 1% (IIc), respectively (Fig. 3). Shrimps, which exhibited very low relative biomass in the first time period, exhibited increases in relative biomass to about 12% in IIa before declining to around 5% in IIb and IIc. Also mostly absent in the first period, Greenland halibut (*Reinhardtius hippoglossoides*) and Atlantic halibut became more present in the catch, with Greenland halibut reaching 4% in subperiod IIb before decreasing to 2% in IIc, whereas the relative biomass of Atlantic halibut was fairly stable around 2% with a maximum in 2016 at 4%. Finally, Snow crab maintained a mean value of 5% throughout this period.

**Biomass fluctuation index**
The biomass fluctuation index of the 84 taxa found in at least 10 years of the time series provides a unique perspective of the fluctuation in biomass within each taxon through time (Fig. 4). It illuminates when the biomass of a species was higher and lower, with respect to itself throughout the time series. Similar to the previous CONISS clustering analysis (Fig. 3), most species fall into three categories: i) species were present at low amounts throughout the time series; ii) species were experienced their highest biomass in the first half of the time series and at low levels in the second half; or iii) vice versa with low amounts in the first half and high amounts in the

second half of the time series. In the early 1970s to the late 1980s the sGSL ecosystem was primarily dominated by several species that were subject to large commercial fisheries. American plaice, Winter flounder and Thorny skate (*Amblyraja radiata*) all reached maximum biomass in the late 1970s before declining and remaining at low levels, whereas Atlantic cod, White hake, Winter skate (*Leucoraja ocellata*), Redfish, and Haddock (*Melanogrammus aeglefinus*) experienced their highest biomass in the early to mid-1980s before collapsing early in the 1990s (Fig. 4). The 1990s and early 2000s were characterized by the highest number of species captured during the survey. However, as presented in section 3.1, despite the high abundance values of some taxa observed in this period, no taxa could compensate for the loss of biomass from the previously collapsed demersal species. Nonetheless, this period brought a rapid increase in the biomass of other important species such as Atlantic herring and Greenland halibut that remained at mid-levels of biomass for almost 20 years, alongside other smaller bodied fish including some flounders (e.g., Windowpane flounder (*Scophthalmus aquosus*)) and a high diversity of Sculpins and Shanny. Both of these taxonomic groups were prominent in the community from early 2000s until present, displaying their highest biomass throughout the time series along with the rapid re-occurrence of Redfish and the increase of Atlantic halibut (Fig. 4).

## Discussion
This study aimed to evaluate the temporal trends in trawlable biomass and community structure of marine fishes and crustaceans in the sGSL to better understand shifts in ecosystem state, and to provide a framework that allows researchers to detect future changes in the ecosystem. Through examining 51 years of fishery-independent multispecies survey data, we identified substantial declines in trawlable biomass and shifts in community structure in the sGSL across the time series. The fishery-independent research survey indices indicate that the trawlable biomass within the ecosystem has declined, whilst the average abundance remained relatively stable through time (Fig. 2), suggesting individuals within the fish community have been getting smaller. The decline in biomass appears to be driven largely by reductions in the biomass of commercial and demersal fishes, whereas recent increases in the abundance of fish appear to be linked to increases in the number of non-commercial and demersal fish captured by the bottom-trawl survey. The relative biomass of individuals has also shifted drastically, with a major regime shift occurring in the early 1990s, corresponding with the collapse[29,30] of several demersal fish species along the east coast of North America. Prior to the early 1990s, the marine community was more stable from year to year, whereas following the regime shift, interannual variation in relative biomass and species composition increased substantially (Figs. 3 and 4).

Research from across the globe has indicated that the biomass and size of marine fish have been declining worldwide for decades[1,2,31]. It is evident from this study that similar trends are occurring within the sGSL, as biomass is on the decline and fish appear to be getting smaller. Fishing activity directly contributes to the decline in fish biomass by removing fish from the ecosystem. Fisheries tend to target the large and abundant species, and thus can have dramatic impacts on total biomass[32], particularly when fish stocks collapse. For instance, investigations into the impact of industrialized fisheries have found that community biomass was reduced by an average of 80% within 15 years of heavy fishing activity[32]. Here, we observed that the decline in biomass in the sGSL was largely driven by declines in commercial fish species, many of which are larger demersal species, including but not limited to Atlantic cod, American plaice, Winter flounder, and White hake. Non-commercial species also experienced declines in biomass, albeit less dramatic, from the early 1990s–2010, but have since rebounded.

Nevertheless, the recent increases in biomass of non-commercial species have not compensated for the loss of biomass from species such as Atlantic cod, White hake and American plaice, but have slowed the apparent decline. Additionally, the 1990s were a period marked by the introduction of several moratoriums on fishing including on species such as Atlantic cod[33] and Redfish[34] along with major reductions in total allowable catch of several

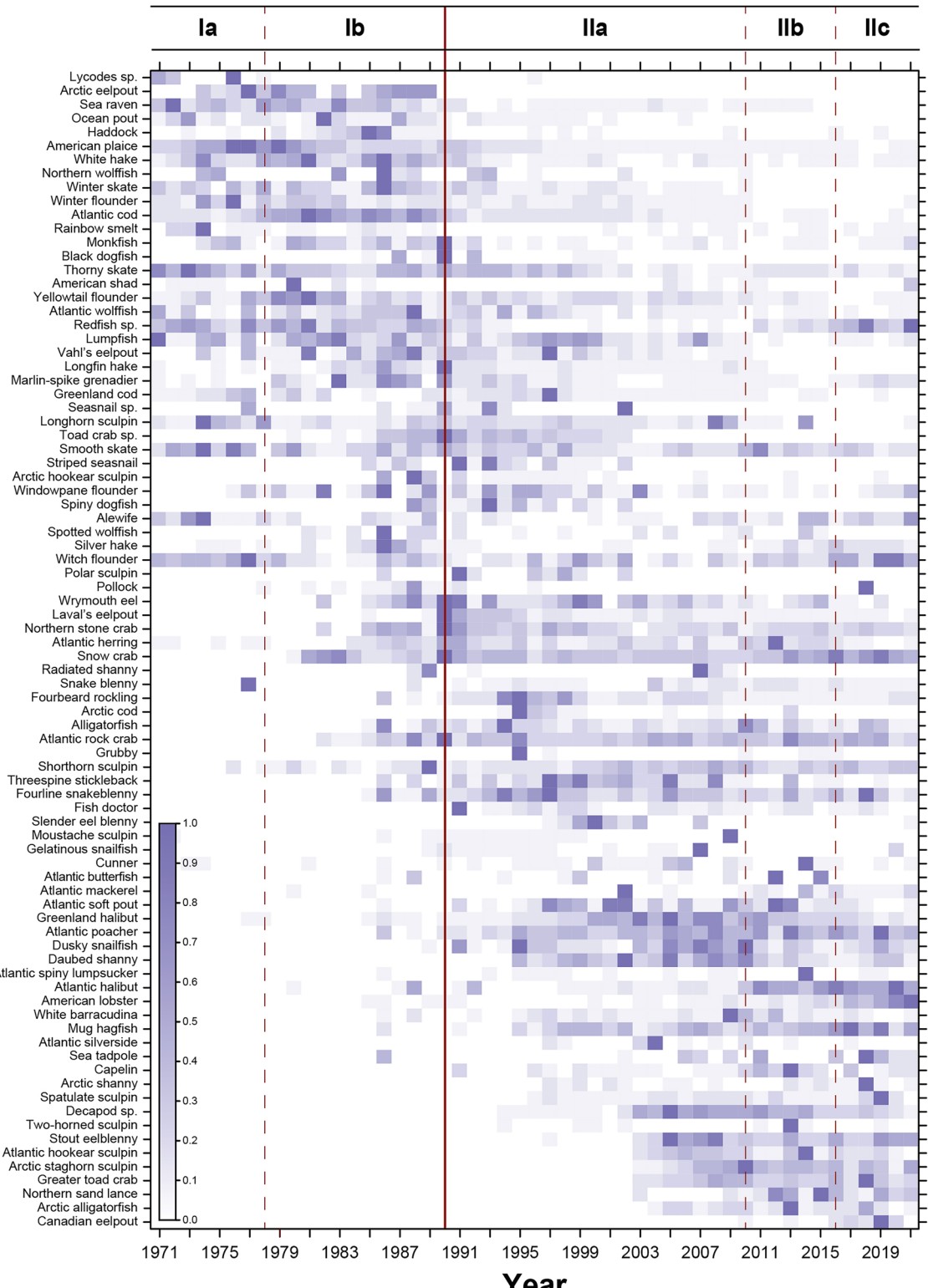

**Fig. 4 | The biomass fluctuation (purple shading) of individual taxa over time.** Individual cells in the table represent a change in taxa biomass relative to their peak abundance, where a value of 1.0 which is the peak biomass is represented by dark purple, and a value of 0 representing lowest relative biomass, is white. The horizontal red line delineates the two main time periods (I and II) identified by the CONISS algorithm and represents a regime shift in the marine community in the sGSL. The header I indicates the first regime time period and II the second regime time period in the fish community. Three dashed red lines delineate subperiods (Ia, Ib, IIa, IIb and IIc) within the main time periods where the patterns of dominance in community member biomass were found to cluster.

other commercial species. Despite moratoriums and reductions in fishing pressure, many populations of commercial species, with the exception of Redfish[34,35], have failed to rebound or return to previous levels of biomass, and continue to exhibit a declining proportion of biomass captured on the yearly trawl survey.

Overfishing has been identified as the principal cause of the collapse of Atlantic cod and other exploited groundfish in the Northwest Atlantic (e.g. refs. 36,37), likely driving the loss of commercial (and demeral) biomass initiated in the 1990s. However, predation can also directly regulate prey populations and indirectly influence their survival by affecting habitat availability, individual growth, and trophic structure[38]. In the sGSL, Grey seal (*Halichoerus grypus*) has experienced significant increases in abundance over the past several decades[39,40] and their predation has been identified as a significant source of mortality, contributing to the lack of recovery for many groundfish stocks[27,41,42] and Atlantic herring[43,44]. The distribution of Atlantic cod, White hake, and Thorny skate has also been strongly influenced by predation risk from Grey seal, with groundfish shifting into lower-risk areas as predation pressure increased in their traditional habitats[45]. However, this shift appears to come at a cost, for instance, Atlantic cod in deeper waters exhibit poor body condition, likely due to reduced food availability[46,47]. High natural mortality has been identified as the primary factor preventing sGSL Atlantic cod recovery[42,48]. Taken together, these factors suggest that while overfishing caused the initial collapse, ongoing high natural mortality from predation could be a key explanation for the continued lack of biomass recovery in the sGSL over the past several decades. Therefore, it is essential to employ ecosystem-based approaches to continue to evaluate the role of predation and corresponding natural mortality on the lack of recovery of many commercial species in the sGSL and to better understand shifts in fish biomass and community structure.

Interestingly, some species such as American lobster, have experienced increasing biomass, despite substantial fishing pressure[49]. Such trends are likely reflective of increasingly suitable habitat available for certain species as the ecosystem changes. A reduction in overall demersal biomass along with increases in water temperature within the sGSL have been favorable to species such as American lobster by opening up large amounts of niche space[50,51]. Therefore, despite substantial and persistent fishing pressure, the species has been able to thrive, highlighting the fact that as some species face negative shifts in response to ecosystem changes, others can flourish by filling empty niche space and capitalizing on a reduction in predation pressure. However, it is important to note, as the ecosystem continues to change and waters persistently warm, ecosystem conditions may become unfavorable for species that have seen positive increases in biomass and or abundance[51], underscoring the need to continue to evaluate and quantify community-level changes in response to changing ecosystems.

Results indicate that from 1971 to 1990, prior to the drastic declines in trawlable biomass, the sGSL fish community was fairly consistent and stable, with low turnover in taxa (Fig. 3). During this time Atlantic cod was an ubiquitous species and was encountered in high densities in every region of the sGSL[33]. In addition to Atlantic cod, flat fish species such as American plaice and Winter flounder were fairly dominant in the fish community. Additionally, Redfish also had a significant presence in the fish community. Beginning in 1978 Atlantic cod became even more dominant with relative abundances fluctuating around 50–60%, which coincided with reductions in the relative abundance of the other dominant demersal fish species (American plaice, Winter flounder and Redfish). Such shifts in relative abundance, but not in species composition, could be the result of complex interspecific interactions in conjunction with ecosystem conditions[52] that benefited Atlantic cod. However, after Atlantic cod experienced a fisheries-induced collapse in the early 1990s, there was a major shift in fish community structure, with an increase in the diversity of taxa captured and species such as Atlantic herring exhibited higher relative abundances that fluctuated until about 2016, when relative abundance of the species declined rapidly as American lobster became more dominant. This finding has some similarities to results from research conducted in the northern Gulf of St. Lawrence which suggested a fishing induced reduction in large

commercially exploited demersal fishes resulted in a regime shift of species composition to a more pelagic based fish community[23]. Concurring observations have been made on Canada's Scotian Shelf, where the over-fishing of large-bodied demersal fishes and their subsequent population collapses resulted in an increase of planktivorous, forage fish species and macroinvertebrates[53], although more recent evidence suggests a return path towards benthic fish species domination, highlighting the transient and resilient nature of this ecosystem[54]. The trends observed in our study of the sGSL are unique as we continue to observe a net loss of trawlable biomass over 30 years despite the increased abundance and biomass of some selected species, suggesting a lack of overall recovery in the system that is important to observe moving forward.

The substantial decline in trawlable biomass observed in the sGSL aligns with the characteristics of a regime shift, where ecosystems transition from one state to another in response to external pressures, with no return to the previous state, even if those pressures are reduced or removed[55]. A pronounced shift in the sGSL community structure began in the early 1990s, marked by the collapse of large demersal groundfish (e.g., Atlantic cod and American plaice) and followed by limited recovery despite fishing moratoriums and reductions in harvest pressure. Concurrently, the sGSL has experienced environmental changes, including warmer waters, a shorter ice season, lower ice volume[56], shifts in primary and secondary production and phenology, and a transition from cold-water to warm-water copepod species[57]. A regime shift analysis revealed an abrupt transition in both sea surface temperature and environmentally-driven spring spawning Atlantic herring recruitment in the early 1990s, shifting from a cold-water/high-recruitment regime (1978–1991) to a warmer-water/low-recruitment regime (1992–2017)[58,59]. This pattern is consistent with observations of regime shifts in the North Atlantic[60], where climate variability, fishing pressure, and trophic interactions collectively drove ecosystem reorganization.

Regime shifts in primary production components have been documented in adjacent ecosystems, influencing fish recruitment and broader food web dynamics. In the Northeast US Continental Shelf, for example, shifts in recruitment success of many fish species broadly coincided with changes in the copepod community[61]. Shifts in marine fish productivity are common, as a study of 230 fish stocks by Vert-Pre, K. et al.[62] showed that 69% were best explained by models incorporating productivity shifts, highlighting their prevalence across marine ecosystems. However, unlike many documented regime shifts where declines in large predatory fish lead to compensatory increases in small pelagic fish or invertebrates[23,53], the sGSL experienced a net loss of trawlable biomass over three decades despite reduced harvest pressure, suggesting a departure from expected resilience mechanisms. The increased turnover in community composition post-1990s provides further evidence of a regime shift possibly driven by both climate forcing and anthropogenic exploitation[60]. These findings underscore the need for continued monitoring and adaptive management strategies that acknowledge the potential irreversibility of large-scale ecological changes in the sGSL.

The post-1990s have been characterized by a period of higher turnover, substantial changes in the relative biomass of community members, and an increase in the presence of non-commercial fish species such as Shanny and Sculpins, which could be influenced by changing physical and biological ecosystem conditions coupled with reduced abundance of previously common commercial fish species. A previous study identified drastic changes in fish community structure in the sGSL, with a shift toward smaller fishes as a result of fishing pressure and high levels of top-down predation control by seals, along with climatic conditions[28]. The reduced presence of collapsed commercial species appeared to have opened up habitat niche space and altered predation pressure on smaller non-commercial fish, which could have led to increases in abundance and/or increases in availability to the trawl gear. It is important to note that our study cannot discern whether there are actually more individuals of non-commercial species, or if individuals have simply altered habitat occupancy and become more available to the trawl, or a combination of both. Nevertheless, similar regime shifts in marine community structure have been observed across the

Northern Hemisphere as a result of a combination of changes in climate and overfishing[63] and are expected to continue to occur in the future.

It is necessary to highlight that shifting baselines also applies at time scales that exceed the generational period discussed in ref.[13]. This is an important consideration since marine ecosystems were not in a pristine state prior to the onset of scientific trawl surveys. The period of time covered by the surveys must be contextualized in light of the commercial fisheries that already existed before surveys started[48]. Nevertheless, the 51-year time series examined in this study is still subject to shifting baseline syndrome, as several generations (i.e. minimum of two generations) of fisheries scientists have traversed this time period. It remains that scientific trawl surveys are often the fisheries-independent source of information covering the longest time period, remain highly effective to support ecological analyses, and can help buffer against shifting baseline syndrome when the full time series available is considered. Our study also takes place in September each year, this fixed timing has benefits for comparability among years, but also has drawbacks in light of shifting climate. Climate change has altered the phenology, migration timing and occupancy of some species[64], potentially impacting our perception of their abundance and biomass within the sGSL during the month of September through time. However, most species that undertake important migrations do so at a later time of the year and in many cases, within the confines of area covered by the survey. Moreover, the survey has been able to track shifts in distribution of species over time in the sGSL system (e.g., ref.[65]). As we are looking at trends across taxa within the confines of the sGSL ecosystem, this effect should not have substantial impacts on overall trends, and our study still provides a robust examination of changes in biomass of the community examined specifically within the month of September. Furthermore, our study is only able to draw insight on shifts in biomass and community structure for taxa susceptible to capture by bottom trawl, thus, other important marine species such as sharks, seals, whales, tuna, swordfish may have undergone changes in biomass but cannot be considered here. The bottom trawl gear used is also more effective at catching certain taxa (e.g. demersal fishes) compared to others (e.g. pelagic fishes), therefore the true biomass and abundance of taxa less susceptible to the bottom trawl are less reliably tracked by the survey, and they could be increasing or decreasing more than observed in the analysis. As such, trends in biomass observed in this study can be impacted by relative catchability to the trawl gear and might not be fully representative of what is occurring in the system. Nevertheless, overall throughout the time series, this susceptibility should be relatively stable, thus offering insight into trends in biomass and community structure in a consistent manner. Inclusion of catchability coefficients for all 122 taxa examined was avoided in this case as some taxa would have extremely poorly informed catchability estimates, increasing the potential to skew trends and interpretation across the time series. Rather, by assuming susceptibility to the trawl remains relatively stable for taxa through time, the survey is able to provide relative changes in biomass and community structure for taxa captured by the fishing gear.

Here we identify and articulate substantial changes in trawlable biomass and community structure within the sGSL ecosystem over the course of 51 years, providing a foundation to evaluate future changes in the marine ecosystem. Shifts in marine community structure have occurred and are predicted to continue to occur in the coming decades according to anticipated climate change (see modeling work from ref.[66]), which will alter thermal regimes, prey availability and habitat suitability. Furthermore, changes in fishing restrictions and fishing pressure in conjunction with increases in top predator abundance and predation rates may continue to have profound impacts on the biomass and community structure within the sGSL and marine systems worldwide. Therefore, it is essential to continue to monitor and evaluate changes in marine ecosystems using fishery-independent surveys as we traverse the Anthropocene.

## Methods
### Data collection
This study focuses on the sGSL, which is a shallow semi-enclosed sea that covers an area of approximately 85,000 km² south of the deep Laurentian

Channel and is bordered by four Canadian provinces, Quebec, New Brunswick, Nova Scotia and Prince Edward Island (Fig. 1). The abundance and biomass of the taxa, and the community structure of the marine ecosystem throughout the sGSL were determined using the data collected during the September multi-species bottom-trawl survey conducted by the DFO since 1971. This survey follows a stratified random sampling design, which covers an area of 73,214 km² and includes sampling of fish and invertebrates using a bottom trawl (Fig. 1). The bottom trawl does not capture all organisms with equal efficiency, but overall, the trawl survey was designed to provide relative biomass and abundance trends for fish and invertebrates distributed between depths of about 20–350 m. The same stratification scheme has been used since 1971, with the exception of the addition of three inshore strata (401–403) in 1984. The analyses are presented here for the 24 strata (415–439) sampled since 1971, representing an area of 70,091 km². The survey indices have been standardized for changes in survey vessels, gears, and protocols which have occurred over the time series using comparative fishing between vessels and gears[17,67]. Stratified random survey indices are expected to be proportional to biomass and abundance for most species.

### Data analyses
For the purpose of this study, we used 122 fish and crustacean taxa that were captured, identified and processed for total catch weight in the selected strata (Supplementary Table 1). Species were categorized based on their expected zonal vertical position within this marine ecosystem (demersal vs pelagic species), and if they are or were under a commercial fishery throughout the time series. Crustaceans included in the analysis consisted of several crab species, American lobster, and several decapod shrimp and prawn species (hereafter referred to as shrimps). Shrimps were analyzed as a group as species identification was inconsistent across the time series. Additionally, the temporal range of shrimp data was limited to start in 1980 as these taxa were unreliably recorded prior to that year.

All analyses were conducted in the software environments C2[68] and R version 4.2.2[69], and the package "rioja"[70] was used for cluster analyses. The data used in this paper are available on the Government of Canada Open Data Portal[71].

#### Grouped biomass and abundance indices. For each species group (i.e., all taxa, demersal, pelagic, commercial, non-commercial, crabs, shrimps, and lobster), the stratified mean estimates of total biomass expressed in standardized kg per tow and total abundance expressed in standardized number per tow were computed[72,73]. The total biomass and abundance indices for each group (i.e., all species, demersal, pelagic, commercial, non-commercial, crabs, shrimps, and lobster) were first plotted across the time series to visualize and compare changes in the biomass indices related to the biomass and abundance of each group through time.

#### Relative biomass index. The relative biomass of each taxon in the marine ecosystem was computed from their respective yearly biomass (kg) relative to the total biomass in that year across taxa in order to evaluate the fluctuation in proportions of taxon biomass through time. The taxa with at least 3.4% relative biomass in any given year over the entire time series were then plotted on a vertical chronological diagram along with the results of a Detrended correspondence analysis (DCA), using the 'rioja' package, which was used to estimate the compositional turnover (i.e. beta-diversity)[74]. This compositional turnover quantifies the magnitude of change in the composition of taxa (biomass and membership) from year to year (i.e. dissimilarity from year to year), where higher values represent larger changes in community composition from one year to the next. The DCA was conducted using 26 segments, which is the default number of segments in the decorana() function in the 'rioja' package. Finally, the 'rioja' package was also used to conduct a constrained hierarchical clustering using a CONISS algorithm (broken-stick model) as a numerical zonation to evaluate when the most important turnover in taxa happened during the time series[75]. This approach is

commonly applied it ecology to evaluate potential regime shifts in multivariate data[76], as such we apply it here.

**Biomass fluctuation index.** Taxa biomass fluctuation over time was estimated using the same stratified mean estimates as previously described (section Grouped Biomass and Abundance Indices, above) but this time applied to individual taxa. Only the taxa found in at least 10 years over the time series were considered for this analysis, which reduced the number of taxa to 84. As previously described in Benoît & Swain[28], for all selected taxa ($i$) and year ($y$), stratified mean ($b_y^i$) were standardized ($B_y^i$) to reduce the values to an interval between 0 (lowest observed biomass over species ($i$) time series) and 1 (highest observed biomass over species ($i$) time series):

$$B_y^i = \frac{b_y^i - min(b^i)}{max(b^i) - min(b^i)}$$

A canonical analysis was conducted using C2 software[68] for the sole purposes of ordering the taxa based on similarity throughout the years. Taxa were ordered on a heatmap using their first axis score of the canonical analysis in order to better depict their biomass fluctuation over time in relation to other species following[28].

### Reporting summary
Further information on research design is available in the Nature Portfolio Reporting Summary linked to this article.

## Data availability
The data used in this paper are all available on the Government of Canada Open Data Portal (https://open.canada.ca/data/en/dataset/1989de32-bc5d-c696-879c-54d422438e64). The data can be accessed by following the link provided. Source data for all figures presented are available in the Supplementary Data 1, 2 and 3 files.

## Code availability
All code used for this study is available on the corresponding author's GitHub (https://github.com/jakeburb/SustainedLossofBiomass). All R code was generated with R version 4.2.2. For any help with the code, please contact the corresponding author.

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

## Acknowledgements
Funding for this work was provided by Fisheries and Oceans Canada. Throughout the years, we are grateful to members that have participated in the September multispecies research vessel survey, which includes DFO employees, volunteers, and students from numerous universities from the east coast of Canada.

## Author contributions
J.L.M. and T.D.T. conceived the idea for the research, N.R. and J.B. performed the research, N.R. and J.B. conducted the analysis, J.B., N.R., J.L.M., F.T., T.D.T., D.R. and F.E.S. contributed to and critically revised the manuscript, J.B. and N.R. wrote the manuscript.

## Competing interests
The authors declare no competing interests.
