## [Transparent Peer Review file · Communications Biology]

Substantial loss of trawlable biomass and lack of recovery in a marine ecosystem

Corresponding Author: Dr Jacob Burbank

Version 0:

Reviewer comments:

Reviewer #1

(Remarks to the Author)

I thank the authors for this very interesting study and important contribution toward our understanding of marine community ecology, in particular in the context of anthropogenic impacts. Their finding of a shift in the fish/macroinvertebrate community of the southern Gulf of St. Lawrence is novel and worthy of publication, although as I discuss below I think the authors might want to use the term 'regime shift' with more care. I especially enjoyed their simple, rather elegant analyses and visualizations. I do have some concerns with the paper, however, related both to the analyses and the way that they are described in the text.

First, I worry about how the authors have treated the data. The study is based on bottom trawl surveys, which are typically designed to target demersal fish. In my experience, such surveys can also reliably catch certain benthic organisms (especially fish) but are ill-suited for targeting pelagic animals. The authors note in lines 434-435 that the trawl was designed for 'fish and invertebrates distributed between 20 m and 350 m' but I am still not sure if it targets all those species equally well. I would be surprised if so; if not, then species-specific catchability factors ought to be included when they make comparisons between species. If the authors believe that catchability factors are unnecessary or impractical, they should provide some justification as to why.

A second, smaller issue related to the dataset is that of timing. My understanding is that the survey always takes place in September, but several of the species analyzed here are migratory. If migration or other phenological patterns have changed over time, it is possible that the community biomass is not changing as dramatically as they report, but rather that the survey simply no longer is timed to capture as much biomass as in the past. I am not convinced that this is an explanation for the trends the authors report, but it is a consideration worth mentioning even briefly in the discussion.

Third, I have concerns with the statistical methods chosen. For regime shift detection, the authors use a chronological clustering approach. Haines et al (2024) recently tested the same approach against simulated data and found very high false positive rates (up to 90%). I encourage the authors to grapple a bit more with the shortcomings of their chosen regime shift detection method, perhaps confirming their results with another approach and/or completing simulation testing to ensure confidence in their results. Additionally, while I do not have personal experience with the DCA, my understanding is that one should report the number of segments used, and that while the first axis reflects species turnover rates, the second axis is not interpretable. However, the authors describe the second axis as reflecting environmental factors in the caption of Figure 3. I worry this may be misleading.

Lastly, and perhaps most importantly, I think the paper would be improved if the authors revise how they frame the study and its key findings. The title asks, "Where have all the fish gone?" but this is not a question that is answered or even addressed by the study itself. Moreover, one of the most interesting results is that all the fish are not in fact gone, since overall abundances have remained constant or even increased, despite observed recent declines in biomass. The second half of the title describes "sustained loss of biomass," suggesting a singular trend, which runs somewhat counter to the idea that the system underwent a regime shift in the early 1990s. This finding of a regime shift, if it can be more robustly confirmed, seems to me to be the most important result. I encourage the authors to focus their framing on the idea of a systemic changepoint, and to engage more deeply with regime shift literature, both as a theory (eg., work of Scheffer and colleagues) and as it relates to the North Atlantic (eg., Beaugrand et al., 2008).

Indeed, the introduction section also seems to ignore some highly relevant literature, especially with claims in lines 105-106 that “studies examining...long-term, high-resolution fishery-independent survey data are lacking.” Such studies do indeed exist. A few examples from the nearby northeast US continental shelf region include: Lucey & Nye (2010), Fenwick et al. (2024), Mills et al. (2024), and Friedland et al. (2023). The latter two references relate to tropicalization and decreases in body size, which relates to the authors’ findings here. I am confident that additional examples exist with other trawl survey datasets (such as those collated in the global FISHGLOB dataset). I do not think the authors need to claim it is novel to analyze survey data in order to justify their work – it is enough that the analysis has not yet been done on the southern Gulf of St. Lawrence data.

A few other small comments:

Line 49: “resulting” seems a poor word choice since they do not formally analyze causation;

Line 158: it is hard to see how this study paves the way for ecosystems that lack long-term monitoring data;

Line 181: “significantly” is a poor word choice if this is not tested statistically;

Line 236: spell out acronym upon first use;

Line 294 (and elsewhere): “collapse” seems a possibly contentious word choice unless a fishery collapse was documented elsewhere, in which case those sources should be cited;

Line 311: “total biomass” is misleading since they only analyze species caught in the trawl survey;

Figure 1: map needs a depth legend;

Figure 2: “All Fish” should be “All Taxa” since it includes crustaceans, and the meaning of shading should be defined in the caption;

Figures 3-4: legends reference “abundance” but I believe biomass-related metrics are being displayed;

I believe that DCA stands for Detrended Correspondence (not Canonical) Analysis;

‘Shifting baseline syndrome’ is mentioned in the introduction but that feels somewhat irrelevant to this study, as the authors rightfully acknowledge in the discussion that surveys started after fishing pressure was already high.

Reviewer #2

(Remarks to the Author)

The manuscript utilize data from an extensive bottom-trawl survey program in the southern Gulf of St Lawrence (1971-2021) to analyse long term changes to community composition, abundance and biomass of >100 crustacean and fish taxa. The authors find that the overall biomass sampled in the survey has declined since the late 1980s, and that species turnover (beta-diversity) has increased. The former is suggested to be due to a combination of reduced mean body size as well as an increase in the abundance of smaller bodied species. A cluster-analysis is also performed, and identify regime shifts in the community.

The MS places itself in a large body of research investigating a previously cod-dominated eastern Canadian ecosystems before and after the collapse and consequential fishing moratorium. However, the fishery independent survey design together with it’s long duration (1971-2021), makes the MS an interesting addition to existing literature. Moreover, I also appreciate the methodological approaches (although I have some questions) used to synthesise the complex multidimensional data material. I believe the MS does a very good job at describing and visualizing the temporal dynamics of the system.

My concerns are mainly towards the framing of the results, specifically with regards to the title and abstract that are only vaguely related to the actual findings. Also, there is a general lack of methodological details and descriptions. Together with confusing wording, the MS results are quite hard to follow, and at times feels like the authors are struggling to. I would also like to see a more detailed results section, perhaps best improved by a more comprehensive appendix.

Below is a near chronological (unprioritized) list of suggestions for improvements.

L1. Title. I have some problem relating the title to the MS. To me, the first part, “Where have all the fish gone?” seem to be related to abundance rather than biomass. I am not a native English speaker but would for example associate the term “all the people” to the number of people, and not their weight. If so, this part of the title may be contradictory to the study findings, that there is a higher number of (smaller) fish now than before. The second part of the title, I believe, is overreaching. The study cannot quantify a general loss in biomass. It’s also hard to see that the “biomass” refers to the fish, if that was the intention. If so, it should be more explicit.

L47. Abstract. The term “whole community biomass” is either wrong or too vague. If by “whole community”, the authors mean the entire ecosystem, this is wrong, as the study cannot capture this based on the available data. However, if the authors mean the community of organisms that can be captured by a bottom trawl, this is true, but not well explained. Is this the (teleost?) fish and crustacean community? There is, also elsewhere in the MS, a mix between using “whole community” and “fish and crustaceans” that could be understood as two different things.

L48 Abstract. I am wondering if the term “species turnover” is not the best wording in the abstract. Is it a well established expression? Intuitively it should relate to the number of how many species are arriving and disappearing in an area. However, does the index beta-diversity (DCCA) quantify this explicitly? A simple redistribution of existing species within a region can lower or increase the beta-diversity. See also later comment.

L142. Introduction. I don’t understand what the sentence part “Given the importance of marine species for overall ecosystem

function [...]” is referring to. Is this the global ecosystem? Surely marine species would be important for marine systems? Also, the remainder of that sentence and the next one (L144-148) are weirdly grandiose concerning how many questions the data can resolve. Is it possible to shorten the sentences and use more moderate formulations?

Figure 2. Should the panels have letters? I guess this goes for all the figures.

Table S-1. Make table searchable. Currently the table is inserted as an image, and it is not possible to search for a species name. Also, there are more letters in the table than are explained in the legend.

Appendix suggestion. The MS present timeseries of 17 species (fig 3), only 15% of the total. Would it be possible to add other species to the appendix? Even though you don't show all 122 taxa, it should be possible to show e.g. the 50-60 most important (in biomass).

Figure suggestion. Alternatively, or additionally, you could make a heatmap like figure 4, but with the relative biomass (from fig 3). This would make it easier to compare the different species' fraction of the total biomass in any given year. This could even be an informative figure in the main text.

L196. When explaining the increasing numbers of demersal specimens per trawl over time, you refer back to the findings from commercial species, where a similar trend was explained by higher numbers of Atlantic herring. What are the small-bodied demersal fish species driving the trend here? This part would also be easier for the reader to look into if more data-plots were made available (see above).

Figure 3. The relative biomass is a useful metric, and I wish it was available for more than the 17 selected species (see above). Also, I don't think the MS has justified or explained why exactly these 17 species were selected.

L239 & L471 DCA. It took a long time before understanding this part. The method description on this is very short. I believe what the authors have undertaken is a Detrended Canonical Correspondence Analysis, traditionally abbreviated as DCCA. The abbreviation used by the authors DCA, most commonly (I think) refer to Detrended Correspondence Analysis, and not Detrended Canonical Analysis. However, I am not certain. In any case, the description of this method is way too brief. Also, the reference to the Birks-article is puzzling. Presumably, this analysis is performed using a R-library or similar. If so, please disclose in the text. A sentence or two about what this index actually represents in biological terms would also be useful. At the moment it is suggested to be that same as beta-diversity. There also seems to be some confusion and overlap between the terms beta-diversity, species turnover (sometimes turnover rate), species composition, compositional turnover, community composition, and community structure.

L240. Here, there is a description of figure 3 and “the first axis of the DCA”, showing “[...] slow decrease in relative biomass”. This is confusing, because in the figure legend, the DCA axis 1 represent “species turnover rates”. I believe the legend is more correct than the MS text.

L285. Here, the text refers to figure 4 regarding the “amplitude of the frequency of turnover within the species composition”. Apart from the sentence being quite convoluted, the authors here suggest that figure 4 (relative biomass) is related to species turnover. This makes no sense to me.

L490. Methods. Please elaborate on what is meant by “first axis score of a canonical analysis”. Is this a new analysis? Is it related to the previous DCA (DCCA)? Is it for a particular year or a mean across all years?

L407. Discussion. Here, the MS touches upon a central point that need to be made much clearer in the MS. The trawl-survey only provides information on organisms actually possible to capture with a bottom trawl. As such, the declining (trawlable) biomass suggest that the survey is failing to sample a gradually growing fraction (biomass) of the ecosystem - assuming that the production is stable. However, the choice of wording throughout the MS give the impression that the survey is somewhat representative for the sGSL ecosystem as a whole. I suggest to perhaps rephrase words like “whole community” as not to mislead the reader.

Perhaps another discussion point is the catchability of non-typical bottom trawl organisms like pelagic fish, lobsters and crabs. Could these organisms be increasing more than the relative trawl-indices suggest?

Anders F. Opdal

Reviewer #3

(Remarks to the Author)

The authors examine trawl survey data from multiple decades in the Southern Gulf of St. Lawrence, and conclude that the species composition of the fish (and invertebrate) community has shifted, that the average amount of biomass caught is less, and they infer that the mean size of organism has gotten smaller. That a loss of fish biomass is observed without any evidence of mass balance, movement/migration, or shift in vital rates is presented as remarkable.

The results of this study are reasonable given the methods used, but the novelty is minimal and essentially this confirms for a relatively small part of Atlantic Canada what has been seen before. Several places around the world have used trawl surveys to show a shift in species composition of a fish community, a shift towards smaller sized fish (which were actually measured and not inferred), and yes, even a loss of biomass. These themes have been documented even in contiguous

northwest Atlantic ecosystems, just 10-15 years ago.

As is, this work is fine, but would be better presented as a NAFO report, an Ecosystem Status Report, in a regional journal, or perhaps in a disciplinary specialty journal like CJFAS or ICES JMS.

The main feature the authors present as quite remarkable is that there has been a loss of biomass observed over time. Which may or may not be the case, but the work is not presented in a compelling manner, the scale and scope of the study may be too small to full track the dynamics of what is occurring, and no sense of hypothesis testing or exploration of where said biomass could have gone is given. The potential exists to truly make this an intriguing piece of work, but much more evidence to lay out the potential sinks of biomass and causality of the loss is needed. Otherwise, as is this is simply a report on observations that again have been demonstrated elsewhere previously.

Version 1:

Reviewer comments:

Reviewer #1

(Remarks to the Author)

Thank you for sharing the revised manuscript - in general, I think it is much improved.

The one issue I still have relates to survey catchability, and I am not quite satisfied with the authors' response. I understand why assuming constant catchability over time makes sense, however, I think that when estimating total biomass as well as when comparing biomass between taxa, it is pretty important to account for catchability. As a simple example, let's assume that 100% of available cod biomass is caught by the survey, but only 60% of herring biomass is. If there is a shift from a cod- to a herring-dominated system (as is the case here), you would perceive a 40% decline in biomass even if there is no actual biomass change.

I don't think that this issue invalidates the results, but I do think the authors should be a little more careful with their claims, and keep the language focused on trawl-available biomass (as they do in the title), rather than talking about net biomass in the community.

Reviewer #2

(Remarks to the Author)

The authors have in an excellent manner improved the MS in accordance with my suggestions. I believe the MS is significantly improved, and is easier to understand.

- Anders F. Opdal

Response to Reviewers Comments

Below, reviewer comments are replicated and *italicized*. Author responses are provided immediately below in non-italicized font. All line numbers refer to line numbers **on the document containing track changes**.

Reviewers' comments:

Reviewer #1 (Remarks to the Author):

I thank the authors for this very interesting study and important contribution toward our understanding of marine community ecology, in particular in the context of anthropogenic impacts. Their finding of a shift in the fish/macroinvertebrate community of the southern Gulf of St. Lawrence is novel and worthy of publication, although as I discuss below I think the authors might want to use the term 'regime shift' with more care. I especially enjoyed their simple, rather elegant analyses and visualizations. I do have some concerns with the paper, however, related both to the analyses and the way that they are described in the text.

Author's Response: We thank the reviewer for their positive and constructive comments that have substantially improve the paper. We address all comments and suggestions in specific responses below.

First, I worry about how the authors have treated the data. The study is based on bottom trawl surveys, which are typically designed to target demersal fish. In my experience, such surveys can also reliably catch certain benthic organisms (especially fish) but are ill-suited for targeting pelagic animals. The authors note in lines 434-435 that the trawl was designed for 'fish and invertebrates distributed between 20 m and 350 m' but I am still not sure if it targets all those species equally well. I would be surprised if so; if not, then species-specific catchability factors ought to be included when they make comparisons between species. If the authors believe that catchability factors are unnecessary or impractical, they should provide some justification as to why.

Author's Response: We thank the reviewer for bringing up this important point. It is true, the bottom-trawl does not capture all fish equally well, it is more effective at catching some compared to others. This is true for all bottom trawl surveys, nevertheless it is very common for bottom trawl surveys to be used to provide fishery-independent indices of biomass and abundance, and these surveys have become a staple of fisheries and ecosystem monitoring programs across Eastern North America and the world (Maureaud et al., 2024). We agree it is important to acknowledge this shortcoming and we have provided justification for not including catchability coefficients in this community level study that looks across 122 taxa. We have now addressed this shortcoming briefly in the methods and expand on it clearly within the discussion.

The altered text in the methods reads "The bottom trawl does not capture all organisms with equal efficiency, but overall the trawl survey was designed to provide relative biomass and abundance trends for fish and invertebrates distributed between depths of about 20 m to 350 m." (Lines 524-526)

The more in-depth added text in the discussion reads "The bottom trawl gear used is also more effective at catching certain taxa (e.g. demersal fishes) compared to others (e.g. pelagic fishes), therefore the true biomass and abundance of taxa less susceptible to the bottom trawl are less reliably tracked by the survey, and they could be increasing or decreasing more than observed in the analysis. Nevertheless, overall throughout the time series, this susceptibility should be relatively stable, thus offering insight

into trends in biomass and community structure in a consistent manner. Inclusion of catchability coefficients for all 122 taxa examined was avoided in this case as some taxa would have extremely poorly informed catchability estimates, increasing the potential to skew trends and interpretation across the time series. Rather, by assuming relative susceptibility to the trawl remains relatively stable for taxa through time, the survey is able to provide relative changes in biomass and community structure for taxa captured by the fishing gear.” (Lines 490-500)

A second, smaller issue related to the dataset is that of timing. My understanding is that the survey always takes place in September, but several of the species analyzed here are migratory. If migration or other phenological patterns have changed over time, it is possible that the community biomass is not changing as dramatically as they report, but rather that the survey simply no longer is timed to capture as much biomass as in the past. I am not convinced that this is an explanation for the trends the authors report, but it is a consideration worth mentioning even briefly in the discussion.

Author’s Response: The reviewer makes an excellent point here and we are glad they have brought this up. We agree, the fixed timing of the survey, along with potential changes in migratory timing of some species over the 51 year dataset associated with changing climate is an important consideration to mention within the discussion. We agree, it is not likely to be the factor explaining the trends in biomass within the ecosystem as most species that undertake important migration in the system do it at a later time and often within the area covered by the survey, but it is an important source of uncertainty that should be explicitly mentioned within the discussion. To address this point we have added the following text to the discussion: “Our study also takes place in September each year, this fixed timing has benefits for comparability among years, but also has drawbacks in light of shifting climate. Climate change has altered the phenology, migration timing and occupancy of some species (Kaimuddin et al., 2016), potentially impacting our perception of their abundance and biomass within the sGSL during the month of September through time. However, most species that undertake important migrations do so at a later time of the year and in many cases, within the confines of area covered by the survey. Moreover, the survey has been able to track shifts in distribution of species over time in the sGSL system (e.g. (Sutton et al., 2024)). As we are looking at trends across taxa within the confines of the sGSL ecosystem this effect should not have substantial impacts on overall trends and our study still provides a robust examination of changes in biomass of the community examined specifically within the month of September.” (Lines 478-487)

Third, I have concerns with the statistical methods chosen. For regime shift detection, the authors use a chronological clustering approach. Haines et al (2024) recently tested the same approach against simulated data and found very high false positive rates (up to 90%). I encourage the authors to grapple a bit more with the shortcomings of their chosen regime shift detection method, perhaps confirming their results with another approach and/or completing simulation testing to ensure confidence in their results.

Author’s Response: We appreciate the reviewer bringing up this point. We have grappled more with the chosen approach for identifying the regime shift. In our case we are examining a regime shift in a multivariate data set, which has multiple species as response variables across multiple years. Therefore, many of the common regime shift approaches are not of use, as they are designed for univariate data sets including methods such as the STARS algorithm, change point approaches and others. When working with multivariate datasets, the primary and widely used approach is a chronological clustering approach. The only other common methods for multivariate data, which based on our reading appears to be less frequently applied is to perform a Principal Components Analysis (PCA) and then implement a

univariate approach such as the STARS algorithm or change point analysis on the first and or second axis of the PCA. Within the paper Haines et al (2024) mentioned by the reviewer, the authors examine 4 of the most commonly used approaches for identifying regime shifts in ecological literature, three methods are designed for univariate data and only 1 method for multivariate data, the chronological clustering approach. The authors of that paper claimed their analysis found all four methods had a similarly high probability of false positive rates. This is one paper, which calls into questions decades of ecological research on regime shifts using well established and widely used methodology. The paper uses a simulation approach, that generates various ecosystem data that is in fact not true data , and generates very similar false positive rates for all four widely used approaches. We feel, this one paper, is not sufficient to deem four widely used regime shift methods that have been published frequently over the years as invalid. With that being said, we did briefly explore two other options to try and detect regime shifts to validate our conclusions of a regime shift in 1990 based on the reviewers recommendation. We conducted a PCA and used the STARS method and changepoint approaches on the first axis scores of the PCA, which detected regime shifts in 1990 and 1988 respectively. An extensive review of regime shifts in ecological research shows that chronological clustering and PCA approaches typically conform, and that they outperform other approaches such as artificial neural network approaches, which are more prone to false positives (Andersen et al., 2009). Therefore, we believe this is sufficient to provide confidence our chosen method in the paper is robust, and provides a realistic identification of the regime shift. Given the regime shift estimate was just a component of our paper, using a well-established method, rather than a paper focused on comparing various methods for identifying regime shifts, there is no need to present all 3 methods examined directly in the paper, as it would detract from the focus of the paper. Furthermore, other recently papers have used less formal methods to identify and claim regime shifts in marine systems such as Mills et al., (2024), which simply examined plots of SST anomalies to deduce a regime shift, and Savenkoff et al (2007), which used inverse modelling to examine changes in consumption, production and biomass of 3 time periods and concluded a regime shift as significant changes were found over time periods. Our DCA axis 1 plot clearly shows a regime shift around 1990, where things go from stable at a low value to increasing, which is then confirmed by the formal chronological clustering method.

Additionally, while I do not have personal experience with the DCA, my understanding is that one should report the number of segments used, and that while the first axis reflects species turnover rates, the second axis is not interpretable. However, the authors describe the second axis as reflecting environmental factors in the caption of Figure 3. I worry this may be misleading.

Author's Response: We agree with the reviewer, we have now reported how many segments were used (n =26 at lines 571). Furthermore, we agree, the second axis is not interpretable, we were trying to highlight it accounts for variation from factors that were unaccounted for in the analysis, but see how the wording could be misinterpreted. We have now adjusted the caption text to simply state the second axis of the DCA represents uninterpretable variation (Lines 817).

Lastly, and perhaps most importantly, I think the paper would be improved if the authors revise how they frame the study and its key findings. The title asks, "Where have all the fish gone?" but this is not a question that is answered or even addressed by the study itself. Moreover, one of the most interesting results is that all the fish are not in fact gone, since overall abundances have remained constant or even increased, despite observed recent declines in biomass. The second half of the title describes "sustained loss of biomass," suggesting a singular trend, which runs somewhat counter to the idea that the system underwent a regime shift in the early 1990s. This finding of a regime shift, if it can be more robustly

confirmed, seems to me to be the most important result. I encourage the authors to focus their framing on the idea of a systemic changepoint, and to engage more deeply with regime shift literature, both as a theory (eg., work of Scheffer and colleagues) and as it relates to the North Atlantic (eg., Beaugrand et al., 2008).

Author's Response: We agree with the reviewer that the title could be changed to align better with the main results of the paper, Reviewer 2 also had suggestions to shift the title and make it more details. In response to both comments we have changed the title to “Substantial loss of trawlable biomass and lack of recovery in a marine ecosystem”

We also thank the reviewer for suggesting to frame finding more within the theoretical framework of a regime shift and engage more with literature from Schaffer and colleagues as well as regime shifts identified within the North Atlantic. We have now added a two paragraphs in the discussion to link our findings with theory and provide comparisons with more literature on the topic in the North Atlantic to round out the framing of our results. The added paragraphs read :

“The substantial decline in biomass observed in the sGSL aligns with the characteristics of a regime shift, where ecosystems transition from one state to another in response to external pressures, with no return to the previous state even if those pressures are reduced or removed⁵⁵. A pronounced shift in the sGSL community structure began in the early 1990s, marked by the collapse of large demersal groundfish (e.g., Atlantic Cod and American Plaice) and followed by limited recovery despite fishing moratoriums and reductions in harvest pressure. Concurrently, the sGSL has experienced environmental changes, including warmer waters, a shorter ice season, lower ice volume⁵⁷, shifts in primary and secondary production phenology, and a transition from cold-water to warm-water copepod species⁵⁸. A regime shift analysis revealed an abrupt transition in both sea surface temperature and environmentally-driven spring spawning Atlantic Herring recruitment in the early 1990s, shifting from a cold-water/high-recruitment regime (1978-1991) to a warmer-water/low-recruitment regime (1992-2017)^{59,60}. This pattern is consistent with observations of regime shifts in the North Atlantic⁵⁶, where climate variability, fishing pressure, and trophic interactions collectively drove ecosystem reorganization.

Regime shifts in primary production components have been documented in adjacent ecosystems, influencing fish recruitment and broader food web dynamics. In the Northeast US Continental Shelf, for example, shifts in recruitment success of many fish species broadly coincided with changes in the copepod community⁶¹. Shifts in marine fish productivity are common, as a study of 230 fish stocks by⁶² showed that 69% were best explained by models incorporating productivity shifts, highlighting their prevalence across marine ecosystems. However, unlike many documented regime shifts where declines in large predatory fish lead to compensatory increases in small pelagic fish or invertebrates^{23,53}, the sGSL experienced a net loss of biomass over three decades despite reduced harvest pressure, suggesting a departure from expected resilience mechanisms. The increased turnover in community composition post-1990s provides further evidence of a regime shift possibly driven by both climate forcing and anthropogenic exploitation⁵⁶. These findings underscore the need for continued monitoring and adaptive management strategies that acknowledge the potential irreversibility of large-scale ecological changes in the sGSL.” (Lines 428-453)

Indeed, the introduction section also seems to ignore some highly relevant literature, especially with claims in lines 105-106 that “studies examining...long-term, high-resolution fishery-independent survey data are lacking.” Such studies do indeed exist. A few examples from the nearby northeast US continental shelf region include: Lucey & Nye (2010), Fenwick et al. (2024), Mills et al. (2024), and

Friedland et al. (2023). The latter two references relate to tropicalization and decreases in body size, which relates to the authors' findings here. I am confident that additional examples exist with other trawl survey datasets (such as those collated in the global FISHGLOB dataset). I do not think the authors need to claim it is novel to analyze survey data in order to justify their work – it is enough that the analysis has not yet been done on the southern Gulf of St. Lawrence data.

Author's Response: The reviewer makes a good point, we appreciate their insight and suggestion for literature to include and agree with the slight reframing of justification. We have now adjusted the section in the intro to acknowledge important work has been done with fishery independent survey data to document and conceptualize ecosystem change through identification of changes in biomass and marine community composition. We now reference some of the studies suggested by the reviewer and highlight these types of analyses are currently lacking for the southern Gulf of St. Lawrence. The adjusted text now reads “Studies examining loss of biomass or shifts in structure of marine communities, including non-commercial species, using long-term, high-resolution fishery-independent survey data have been valuable in conceptualizing ecosystem change along the northeast US continental shelf ¹⁰⁻¹² but are lacking for the southern Gulf of St. Lawrence (sGSL).” (Lines 105-108)

21 12A few other small comments:

Line 49: “resulting” seems a poor word choice since they do not formally analyze causation;

Author's Response: We agree, we removed the word ‘resulting’ and replaced it with ‘and’. Reviewer 2 also provided suggestions for this sentence, so the sentence now reads “ Survey data indicate a substantial decline in community biomass and increase in turnover for taxa susceptible to bottom-trawl fishing gear in the southern Gulf of St. Lawrence marine ecosystem that corresponds with the loss of several large predatory fish and a major regime shift around the early 1990's.”(Lines 46-50)

Line 158: it is hard to see how this study paves the way for ecosystems that lack long-term monitoring data;

Author's Response: This is a good point, we have clarified this paves the way for similar research in ecosystems that currently have long-term bottom-trawl survey data (Lines 160-161).

Line 181: “significantly” is a poor word choice if this is not tested statistically;

Author's Response: We agree, we have changed ‘significantly’ to ‘substantially’ (Line 184).

Line 236: spell out acronym upon first use;

Author's Response: We have now spelled out the acronym (Lines 243-244).

Line 294 (and elsewhere): “collapse” seems a possibly contentious word choice unless a fishery collapse was documented elsewhere, in which case those sources should be cited;

Author's Response: The collapse of groundfish in the southern Gulf of St. Lawrence has been well documented, therefore the use of the word collapse is suitable in this case. We have added two notable references that have documented and examined the collapse of groundfish stocks in the southern Gulf of St Lawrence (Morissette et al., 2009; Savenkoff, Swain, et al., 2007) (Line 326).

Line 311: “total biomass” is misleading since they only analyze species caught in the trawl survey;

Author’s Response: We understand the reviewers concern. Therefore within the discussion we have removed reference to ‘total biomass’ and just say ‘biomass’.. This makes it clear that we are referring to biomass of species the survey can sample, rather than total biomass in the entire community.

Figure 1: map needs a depth legend;

Author’s Response: We agree, we have now readjusted Figure 1 to show the average depth of each statum and added a depth legend (Line 805).

Figure 2: “All Fish” should be “All Taxa” since it includes crustaceans, and the meaning of shading should be defined in the caption;

Author’s Response: We have now changed all fish to all taxa, we thank the reviewer for pointing this out. We have also now defined the shading in the figure caption, this text reads “The shading indicates the 95 % confidence interval of biomass and abundance estimates.” (Lines 812-813)

Figures 3-4: legends reference “abundance” but I believe biomass-related metrics are being displayed; I believe that DCA stands for Detrended Correspondence (not Canonical) Analysis; ‘Shifting baseline syndrome’ is mentioned in the introduction but that feels somewhat irrelevant to this study, as the authors rightfully acknowledge in the discussion that surveys started after fishing pressure was already high.

Author’s Response: We thank the reviewer for pointing this out, they are correct. We have removed all reference to abundance in the captions of Figures 3 and 4 and ensure they say biomass instead. We have also indicated the DCA is representing Detrended Correspondence Analysis. Lastly, we believe that although the surveys started before the onset of high fishing pressure, ‘Shifting Baseline Syndrome’ is still extremely relevant for this study. Given the drastic declines in biomass observed over the time series, it is very relevant to discuss Shifting Baseline Syndromes, as several generations of fisheries scientists have occurred over the 51 years. The baseline scientists had in 1971 are drastically different then the baseline a newer fisheries scientist would have beginning in the 2000s or 2010s. As biomass shifts fisheries scientists are subject to bias and may fall into the trap of referencing biomass relative to when they started their careers, rather than historic biomass, this is one of the main points of the ‘Shifting Baseline Syndrome’. We maintain it is relevant for this study and is an important and valuable concept to introduce, whilst still acknowledging, the time series has its limits and begins when fishing pressure was already high.

Reviewer #2 (Remarks to the Author):

The manuscript utilize data from an extensive bottom-trawl survey program in the southern Gulf of St Lawrence (1971-2021) to analyse long term changes to community composition, abundance and biomass of >100 crustacean and fish taxa. The authors find that the overall biomass sampled in the survey has declined since the late 1980s, and that species turnover (beta-diversity) has increased. The former is suggested to by due to a combination of reduced mean body size as well as an increase in the abundance of smaller bodied species. A cluster-analysis is also performed, and identify regime shifts in the community.

The MS places itself in a large body of research investigating a previously cod-dominated eastern Canadian ecosystems before and after the collapse and consequential fishing moratorium. However, the fishery independent survey design together with it’s long duration (1971-2021), makes the MS an interesting addition to existing literature. Moreover, I also appreciate the methodological approaches (although I have some questions) used to synthesise the complex multidimensional data material. I believe the MS does a very good job at describing and visualizing the temporal dynamics of the system.

Author’s Response: We thank the reviewer very much for their positive and constructive comments, which have improved the manuscript. We have addressed all specific comments below.

My concerns are mainly towards the framing of the results, specifically with regards to the title and abstract that are only vaguely related to the actual findings. Also, there is a general lack of methodological details and descriptions. Together with confusing wording, the MS results are quite hard

to follow, and at times feels like the authors are struggling to. I would also like to see a more detailed results section, perhaps best improved by a more comprehensive appendix.

Author's Response: Following this and Reviewer 1's recommendation we have now changed the title of the manuscript to "Substantial loss of trawlable biomass and lack of recovery in a marine ecosystem". We also have ensured the abstract is directly reflective of our findings and are confident it represents the main results of the work. We have gone through the methods section and improved the level of detail, making sure sufficient detail is included so others can reproduce the study. We have also carefully combed through the results and have ensured they are clear and follow a logical order. We have addressed recommendations for further results in an added figure in the appendix. Additional details can be found in responses to the detailed comments below, where each concern is thoroughly addressed.

Below is a near chronological (unprioritized) list of suggestions for improvements.

L1. Title. I have some problem relating the title to the MS. To me, the first part, "Where have all the fish gone?" seem to be related to abundance rather than biomass. I am not a native English speaker but would for example associate the term "all the people" to the number of people, and not their weight. If so, this part of the title may be contradictory to the study findings, that there is a higher number of (smaller) fish now than before. The second part of the title, I believe, is overreaching. The study cannot quantify a general loss in biomass. It's also hard to see that the "biomass" refers to the fish, if that was the intention. If so, it should be more explicit.

Author's Response: We thank the reviewer for their suggestion to alter the title. Reviewer 1 also suggested the title be changed to omit the first part. In response to both reviewers comments we have changed the title to "Substantial loss of trawlable biomass and lack of recovery in a marine ecosystem".

L47. Abstract. The term "whole community biomass" is either wrong or too vague. If by "whole community", the authors mean the entire ecosystem, this is wrong, as the study cannot capture this based on the available data. However, if the authors mean the community of organisms that can be captured by a bottom trawl, this is true, but not well explained. Is this the (teleost?) fish and crustacean community? There is, also elsewhere in the MS, a mix between using "whole community" and "fish and crustaceans" that could be understood as two different things.

Author's Response: We thank the reviewer for bringing up this point. We agree, 'whole community biomass' is not the appropriate term to use here. We are referring to biomass of fish and crustacean species that are susceptible to capture by the bottom trawl. We have now removed the word 'whole' from the sentence and clarified we are referring to biomass of taxa susceptible to capture by bottom-trawl, the sentence now reads "Survey data indicate a substantial and sustained decline in community biomass and increase in turnover for taxa susceptible to bottom-trawl fishing gear in the southern Gulf of St. Lawrence marine ecosystem that corresponds with the loss of several large predatory fish and a major regime shift around the early 1990's."(Lines 46-50)

L48 Abstract. I am wondering if the term "species turnover" is not the best wording in the abstract. Is it a well established expression? Intuitively it should relate to the number of how many species are arriving

and disappearing in an area. However, does the index beta-diversity (DCCA) quantify this explicitly? A simple redistribution of existing species within a region can lower or increase the beta-diversity. See also later comment.

Author's Response: We maintain 'turnover' is the appropriate term to use in this case and that it is a widely accepted term in ecology. In a very recent paper published in Nature (Pinsky et al., 2025), the authors describe turnover thoroughly, they state within the opening of their introduction that "Turnover occurs when some species increase their abundance or occupancy through time and others decline. Such temporal turnover has marked effects on the structure and functioning of ecological communities and can be rapid even while the number of species remains relatively unchanged". This is what we are referring to here in our analysis. Compositional beta diversity measured by the DCA axis 1 is quantifying changes in community composition, including changes in the relative biomass and occupancy of taxa captured by the survey. Therefore when rates of change of community composition is higher, we see higher 'turnover'. We have removed reference specifically of 'species turnover' and rather state turnover of taxa, since we are talking about taxa and not just species in the analysis.

L142. Introduction. I don't understand what the sentence part "Given the importance of marine species for overall ecosystem function [...]" is referring to. Is this the global ecosystem? Surely marine species would be important for marine systems? Also, the remainder of that sentence and the next one (L144-148) are weirdly grandiose concerning how many questions the data can resolve. Is it possible to shorten the sentences and use more moderate formulations?

Author's Response: We thank the reviewer for this comment. We agree, these sentences could be shorted and clarified to present a more direct and relevant message, better linking to the scope of the paper. We have now altered the text to read ". Given the complexity of interspecies interactions, it is valuable to use available data from fishery-independent multispecies research surveys to examine and quantify changes in biomass and community structure through time to better understand potential impacts of anthropogenic activity and climate change on marine ecosystems. It is only when community-level shifts, rather than only species specific shifts are examined, that changes in ecosystem over time can be better articulated.

Figure 2. Should the panels have letters? I guess this goes for all the figures.

Author's Response: We choose to give each panel a clear, descriptive title, therefore omitting the need for lettering. If the editor required numbers to be added to the panels we will gladly add them, but for now we believe the descriptive titles for each panel provide clear distinction for each panel.

Table S-1. Make table searchable. Currently the table is inserted as an image, and it is not possible to search for a specie name. Also, there are more letters in the table than are explained in the legend.

Author's Response: We have now updated the Supplementary materials to make the table searchable. We also made sure there is only D (representing demersal) and P (representing pelagic) within the table. The previous version also had a code B (representing benthic), that should have been represented as D. We thank the reviewer for catching this error.

Appendix suggestion. The MS present timeseries of 17 species (fig 3), only 15% of the total. Would it be possible to add other species to the appendix? Even though you don't show all 122 taxa, it should be possible to show e.g. the 50-60 most important (in biomass).

Author's Response: We understand the reviewers request and do agree it is a good idea to provide relative biomass plots for additional species in the supplementary material. Within the paper we show the 17 taxa with the highest relative biomass throughout the time series. We used relative biomass cutoff of 3.4% throughout the time series, thus taxa with at least 3.4% relative biomass in any given year across the time series are included in Figure 3 of the main paper. This allows readers to see clearly the most important species in terms of relative biomass. We now have provided an additional plot in the supplementary material (Figure S1) that shows the relative biomass across the time series for an additional 23 taxa. These taxa represent taxa that had a relative biomass of between 0.17 and 3.39% in at least one year across the time series. The reviewer originally suggested showing 60-70 more species but we believe this is of limited value given the remaining species have very low relative abundances. We believe the addition of these extra 23 taxa provides the reader with more relevant insight into changes in relative biomass. The new Figure S-1 is as follows:

Figure suggestion. Alternatively, or additionally, you could make a heatmap like figure 4, but with the relative biomass (from fig 3). This would make it easier to compare the different species' fraction of the total biomass in any given year. This could even be an informative figure in the main text.

Author's Response: We thank the reviewer for this suggestion but believe a figure of this nature would not be beneficial to the paper and would provide too much visual overlap with Figure 4. Additionally, many of the species have very low relative biomass throughout the time series, so for most species the heat map would not be very informative when scaled from 0-1. As the reviewer suggested we added a

figure in the supplementary material to show the relative biomass of additional species for interested readers.

L196. When explaining the increasing numbers of demersal specimens per trawl over time, you refer back to the findings from commercial species, where a similar trend was explained by higher numbers of Atlantic herring. What are the small-bodied demersal fish species driving the trend here? This part would also be easier for the reader to look into if more data-plots were made available (see above).

Author's Response: Here in the text we indicate the number of specimens per tow (abundance) over the last decade has a similar trend to the abundance of non-commercial species (not commercial species). This increase in abundance is linked to increase in both commercial and non-commercial demersal species, but shows a similar trend to the non-commercial species. The increase in abundance for demersal species is linked to the increase in numbers of a wide range of non-commercial species, too many to list within the results. We simply pointed out Atlantic Herring and Redfish spp. in the commercial fish trends, as they are two species driving those recent trends and it was imperative to clarify although Atlantic Herring abundance in the survey increased in 2010-2014, it since declined . Given we are looking here at the broad trends across demersal species, it is not possible to single out a few non-commercial species driving this similarity between the two groups, it is more that several non-commercial species are also demersal, as can be seen on the species list. Overall, relative biomass plots do not help decipher which non-commercial taxa are driving this trend, as they only show relative biomass, not relative abundance (numbers of fish).

Figure 3. The relative biomass is a useful metric, and I wish it was available for more than the 17 selected species (see above). Also, I don't think the MS has justified or explained why exactly these 17 species where selected.

Author's Response: Thank you for this point, we agree the figure is very valuable to help understand patterns in relative biomass through time and to help visualize the regime shift. As per the reviewer suggestion we added a supplementary figure showing relative biomass of 23 additional species. We do provide clear justification for inclusion of the 17 species, the text reads "The taxa with at least 3.4% relative biomass in any given year over the entire time series were then plotted on a vertical chronological diagram along with the results of a Detrended correspondence analysis (DCA), using the 'rioja' package, which was used to estimate the compositional turnover (i.e. beta-diversity) ⁷⁴"

L239 & L471 DCA. It took a long time before understanding this part. The method description on this is very short. I believe what the authors have undertaken is a Detrended Canonical Correspondence Analysis, traditionally abbreviated as DCCA. The abbreviation used by the authors DCA, most commonly (I think) refer to Detrended Correspondence Analysis, and not Detrended Canonical Analysis. However, I am not certain. In any case, the description of this method is way too brief. Also, the reference to the Birks-article is puzzling. Presumably, this analysis is performed using a R-library or similar. If so, please disclose in the text. A sentence or two about what this index actually represents in biological terms would also be useful. At the moment it is suggested to be that same as beta-diversity. There also seems to be some confusion and overlap between the terms beta-diversity, species turnover (sometimes turnover rate), species composition, compositional turnover, community composition, and community structure.

Author's Response: We thank the reviewer for the comment. We have infact undertaken a Detrended Correspondence Analysis (DCA), not a DCCA. The reviewer is correct, we did use a R package, 'roiija' to conduct the cluster analysis and DCA, this is already clearly mentioned in the text in the opening section of the methods (Lines 546). We do however agree, the reader should see mention of the use of the package when the DCA and cluster analysis is described in the methods, therefore we have reiterated the R package was used for the analysis in the appropriate section (Lines 567). This approach does provide an indication of compositional turnover (i.e. beta diversity). In our case we are quantifying turnover of taxa within the community (community of taxa susceptible to the bottom trawl). There is rightfully overlap between the terms turnover, and community composition, as we are evaluating turnover in community composition. The Birks article is referenced as it presents and provides a practical application of the compositional turnover metric, although it is a paleoecology study, the approach is equally relevant to our analysis. We have gone through the text and removed reference to 'species turnover' specifically, as we are looking at taxa not just species. As suggested, additional text in the methods was added to provide more concrete and comprehensive description of the DCA and cluster analysis so readers can clearly understand what had been conducted.

The added text now reads "The relative biomass of each taxon in the marine ecosystem was computed from their respective yearly biomass (kg) relative to the total biomass in that year across taxa in order to evaluate the fluctuation in proportions of taxa biomass through time. The taxa with at least 3.4% relative biomass in any given year over the entire time series were then plotted on a vertical chronological diagram along with the results of a Detrended correspondence analysis (DCA), using the 'roiija' package, which was used to estimate the compositional turnover (i.e. beta-diversity) (Birks, 2007). This compositional turnover quantifies the magnitude of change in the composition of taxa (biomass and membership) from year to year (i.e. dissimilarity from year to year), where higher values represent larger changes in community composition from one year to the next. Finally, the 'roiija' package was also used to conduct a constrained hierarchical clustering using a CONISS algorithm (broken-stick model) as a numerical zonation to evaluate when the most important turnover in taxa happened during the time series (Legendre & Birks, 2012). This approach is commonly applied it ecology to evaluate potential regime shifts in multivariate data (Andersen et al., 2009), as such we apply it here." (Lines 562-575).

L240. Here, there is a description of figure 3 and "the first axis of the DCA", showing "[...] slow decrease in relative biomass". This is confusing, because in the figure legend, the DCA axis 1 represent "species turnover rates". I believe the legend is more correct than the MS text.

Author's Response: We agree and appreciate the reviewer pointing this out. We have replaced 'slow decrease in relative biomass' with 'low turnover rates' as the first axis of the DCA was indicating low turnover rates in the period of 1971-1990 (Line 244).

L285. Here, the text refers to figure 4 regarding the "amplitude of the frequency of turnover within the species composition". Apart from the sentence being quite convoluted, the authors here suggest that figure 4 (relative biomass) is related to species turnover. This makes no sense to me.

Author's Response: We thank the reviewer for bringing this point to light. We agree, the sentence was convoluted and needed to be adjusted. We have now adjusted the text to be more clear and accurate. The two opening sentences now reads "The biomass fluctuation index of the 84 taxa found in at least 10 years of the time series provides a unique perspective of the fluctuation in biomass within each taxa

through time (Figure 4). It illuminates when the biomass of a species was higher and lower, with respect to itself throughout the time series.” (Lines 289-292)

L490. Methods. Please elaborate on what is meant by “first axis score of a canonical analysis”. Is this a new analysis? Is it related to the previous DCA (DCCA)? Is it for a particular year or a mean across all years?

Author’s Response: We thank the reviewer for bringing up this confusion. This is a new analysis and this needed to be made more clear in the methods. We have now added text to clear up that a canonical correspondence analysis was done on the biomass fluctuation index for the sole purpose of ordering the species along a heatmap. The added text now reads “A canonical analysis was conducted using C2 software⁶⁸ for the sole purposes of ordering the taxa based on similarity throughout the years. Taxa were ordered on a heatmap using their first axis score of the canonical analysis in order to better depict their biomass fluctuation over time in relation to other species following ²⁸.” (Lines 589-592)

L407. Discussion. Here, the MS touches upon a central point that need to be made much clearer in the MS. The trawl-survey only provides information on organisms actually possible to capture with a bottom trawl. As such, the declining (trawlable) biomass suggest that the survey is failing to sample a gradually growing fraction (biomass) of the ecosystem - assuming that the production is stable. However, the choice of wording throughout the MS give the impression that the survey is somewhat representative for the sGSL ecosystem as a whole. I suggest to perhaps rephrase words like “whole community” as not to mislead the reader.

Author’s Response: We agree the term ‘whole community biomass’ is misleading, we have now removed all mention of ‘whole community biomass’ and make it clear in the abstract, discussion and other section we are referring to community biomass for taxa susceptible to capture by the bottom trawl.

Perhaps another discussion point is the catchability of non-typical bottom trawl organisms like pelagic fish, lobsters and crabs. Could these organisms be increasing more than the relative trawl-indices suggest?

Author’s Response: We thank the reviewer for raising this important point. Reviewer one also had a comment related to the catchability differing across species. As such we have added text in the discussion to explain that catchability/susceptibility is not the same across the taxa considered. We also elaborate on the fact that biomass and abundance of non-typical bottom trawl organisms are not as well tracked by the survey, and thus could be increasing or decreasing more than observed in the analysis . The added text in the discussion now reads “The bottom trawl gear used is also more effective at catching certain taxa (e.g. demersal fishes) compared to others (e.g. pelagic fishes), therefore the true biomass and abundance of taxa less susceptible to the bottom trawl are less reliably tracked by the survey, and they could be increasing or decreasing more than observed in the analysis. Nevertheless, overall throughout the time series this susceptibility should be relatively stable, thus offering insight into trends in biomass and community structure in a consistent manner.” (Lines 491-496)

-Anders F. Opdal

Reviewer #3 (Remarks to the Author):

The authors examine trawl survey data from multiple decades in the Southern Gulf of St. Lawrence, and conclude that the species composition of the fish (and invertebrate) community has shifted, that the average amount of biomass caught is less, and they infer that the mean size of organism has gotten smaller. That a loss of fish biomass is observed without any evidence of mass balance, movement/migration, or shift in vital rates is presented as remarkable. The results of this study are reasonable given the methods used, but the novelty is minimal and essentially this confirms for a relatively small part of Atlantic Canada what has been seen before. Several places around the world have used trawl surveys to show a shift in species composition of a fish community, a shift towards smaller sized fish (which were actually measured and not inferred), and yes, even a loss of biomass. These themes have been documented even in contiguous northwest Atlantic ecosystems, just 10-15 years ago.

Author's Response: We thank the reviewer for taking the time to evaluate our manuscript and for concurring that results are reasonable given the approaches. We understand loss of biomass has been documented elsewhere, but that does not diminish the importance and novelty of this work. In depth examinations of changes in biomass and shift in community composition using the methods presented, and over a timeseries of 51 years is rare, and is novel for the southern Gulf of St. Lawrence ecosystem, an important ecosystem in the North Atlantic that support a wide diversity of species and lucrative fisheries that are imperative for the economy along Atlantic Canada. Therefore examining and articulating trends in biomass and community composition of the marine community is essential to better understand how fish and crustaceans are changing through time. Our work is novel as unlike other studies in nearby areas such as the northern Gulf of St. Lawrence that have seen compensation n biomass from forage fish (Savenkoff et al, 2007), we have seen continued low biomass following the drastic reduction in biomass in the 1990s.

As is, this work is fine, but would be better presented as a NAFO report, an Ecosystem Status Report, in a regional journal, or perhaps in a disciplinary specialty journal like CJFAS or ICES JMS.

Author's Response: We thank the reviewer for confirming they think the work is good as is. We however believe this work is suitable for a higher level journal with broader readership such as Communications Biology. Publishing this work, which will be of interest to the scientific community and general public, in a higher level journal like Communications Biology will allow the work to reach a wider readership and help communicate important trends in a marine ecosystem broadly. These results are important to share widely and for the broad scientific community to be aware of as they highlight the current state of our marine ecosystems compared to historical states. Overall the content and importance of this work is suitable for a journal such as Communications Biology.

The main feature the authors present as quite remarkable is that there has been a loss of biomass observed over time. Which may or may not be the case, but the work is not presented in a compelling manner, the scale and scope of the study may be too small to full track the dynamics of what is occurring, and no sense of hypothesis testing or exploration of where said biomass could have gone is given. The potential exists to truly make this an intriguing piece of work, but much more evidence to lay

out the potential sinks of biomass and causality of the loss is needed. Otherwise, as is this is simply a report on observations that again have been demonstrated elsewhere previously.

Author's Response: We understand the reviewer comment and are very thankful they took the time to review the manuscript but think the review lacks concrete suggestions to improve the paper. However, both Reviewer 1 and 2 have provided concrete, actionable suggestions that have been incorporated and have helped improve the framing and overall content of the manuscript.

We believe the scale and scope of our study is more than sufficient to track the dynamics of the marine ecosystem we are investigating, given the survey implements a reputable stratified random sampling design across the entire sGSL ecosystem and has occurred since 1971.

The main loss of biomass arises from the collapse of commercial groundfish stocks in the 1990s, due to overfishing. Research has then showed that the lack of recovery of these stocks is due to high natural mortality rates induced by grey seal predation. We added a paragraph describing this body of evidence in greater details that reads: "Overfishing has been identified as the principal cause of the collapse of Atlantic Cod and other exploited groundfish in the Northwest Atlantic (e.g., ^{36,37}), likely driving the loss of commercial (and demersal) biomass initiated in the 1990s. However, predation can also directly regulate prey populations and indirectly influence their survival by affecting habitat availability, individual growth, and trophic structure ³⁸. In the sGSL, Grey Seal (*Halichoerus grypus*) have experienced significant increases in abundance over the past several decades ^{39,40} and their predation has been identified as a significant source of mortality, contributing to the lack of recovery for many groundfish stocks ^{27,41,42} and Atlantic Herring ^{43,44}. The distribution of Atlantic Cod, White Hake, and Thorny Skate has also been strongly influenced by predation risk from Grey Seals, with groundfish shifting into lower-risk areas as predation pressure increased in their traditional habitats ⁴⁵. However, this shift appears to come at a cost, as Atlantic Cod in deeper waters exhibit poor body condition, likely due to reduced food availability ^{46,47}. High natural mortality has been identified as the primary factor preventing sGSL Cod recovery ^{42,48}. Taken together, these factors suggest that while overfishing caused the initial collapse, ongoing high natural mortality from predation could be a key explanation for the continued lack of biomass recovery in the sGSL over the past several decades."(Lines 352-366)

It is true some biomass may have moved outside of the system, however some biomass could also have moved into the system and both shifts of species into and out of the ecosystem have implications for the total annual biomass. Even if biomass moved northward out of the ecosystem, this is still a loss of biomass within this unique ecosystem, which has wide ranging ecological and economic repercussions. The potential sinks of biomass outside of the sGSL is outside the scope of this study. We do however now indicate potential shortcomings of our study with respect to shifting distributions in response to reviewer 1s comments by explaining that occupancy and migration patterns of some species may have changed through time due to climate factors, which could impact our perception of biomass. Overall reporting on observations in the ecosystem are essential to help conceptualize the current state of knowledge and set the stage for future work that focuses on causation. With the integration of reviewer 1 and 2s comments we now have improved the framing of our results around regime shift theory, and clearly highlight the novelty of our results, which find a lack of recovery of biomass despite reductions harvest pressure. We strongly believe these finds are worthy of sharing to a wider audience.

Response to Reviewers Comments

Below, reviewer comments are replicated and *italicized*. Author responses are provided immediately below in non-italicized font. All line numbers refer to line numbers **on the document containing track changes**.

Reviewers' comments:

Reviewer #1 (Remarks to the Author):

Thank you for sharing the revised manuscript - in general, I think it is much improved.

The one issue I still have relates to survey catchability, and I am not quite satisfied with the authors' response. I understand why assuming constant catchability over time makes sense, however, I think that when estimating total biomass as well as when comparing biomass between taxa, it is pretty important to account for catchability. As a simple example, let's assume that 100% of available cod biomass is caught by the survey, but only 60% of herring biomass is. If there is a shift from a cod- to a herring-dominated system (as is the case here), you would perceive a 40% decline in biomass even if there is no actual biomass change.

I don't think that this issue invalidates the results, but I do think the authors should be a little more careful with their claims, and keep the language focused on trawl-available biomass (as they do in the title), rather than talking about net biomass in the community.

Authors' Response: We thank the reviewer for bringing up this important point and we agree it is essential to be clear that our work can only speak to trends in trawlable biomass, not biomass of the entire community. Furthermore acknowledging the issues surrounding catchability more clearly in the discussion is warranted. Therefore, we have added the following sentence to make clear the data might not be fully representative of what is going on in the system due to uncertainty surrounding catchability of different species in the survey : "As such, trends in biomass observed in this study can be impacted by relative catchability to the trawl gear and might not be fully representative of what is occurring in the system. " (Lines 489-491)

We also went through the manuscript thoroughly to ensure the focus is kept on 'trawlable biomass'.

Reviewer #2 (Remarks to the Author):

The authors have in an excellent manner improved the MS in accordance with my suggestions. I believe the MS is significantly improved, and is easier to understand.

- Anders F. Opdal

Authors' Response: We thank the reviewer for their comment and for taking the time to review the manuscript.